# Is Limited Participant Diversity Impeding EEG-based Machine Learning?

**Philipp Bomatter**
School of Informatics
University of Edinburgh
`philipp.bomatter@ed.ac.uk`

**Henry Gouk**
School of Informatics
University of Edinburgh
`henry.gouk@ed.ac.uk`

## Abstract

The application of machine learning (ML) to electroencephalography (EEG) has great potential to advance both neuroscientific research and clinical applications. However, the generalisability and robustness of EEG-based ML models often hinge on the amount and diversity of training data. It is common practice to split EEG recordings into small segments, thereby increasing the number of samples substantially compared to the number of individual recordings or participants. We conceptualise this as a multi-level data generation process and investigate the scaling behaviour of model performance with respect to the overall sample size and the participant diversity through large-scale empirical studies. We then use the same framework to investigate the effectiveness of different ML strategies designed to address limited data problems: data augmentations and self-supervised learning. Our findings show that model performance scaling can be severely constrained by participant distribution shifts and provide actionable guidance for data collection and ML research. The code for our experiments is publicly available online.[1]

## 1 Introduction

A century after the first human electroencephalogram was recorded in 1924, Electroencephalography (EEG) is in many ways still the best available technique for the direct and non-invasive measurement of brain activity. In particular, with newer generations of dry-electrode recording hardware, it has a unique set of desirable properties, including its low cost and patient burden, portability, and high temporal resolution. Given the complexity and high dimensionality of EEG signals, machine learning (ML) has shown considerable promise in advancing neuroscientific research and clinical applications [1, 2, 3, 4]. However, the successful deployment of ML in EEG-based applications is not without challenges. A fundamental obstacle is the limited availability of high-quality EEG data, which is often sparse, noisy, and costly to collect.

While prior works have acknowledged limited data as a challenge for EEG-based machine learning [5, 6], systematic studies are currently lacking. In particular, an aspect that has largely been ignored is the hierarchical nature of the data generation process relevant to EEG machine learning work. EEG-based ML models rarely operate on full recordings; samples usually correspond to more manageable segments on the order of a few seconds, extracted from recordings through sliding-window approaches [5]. As a result, the overall sample size is usually considerably larger than the number of individual participants in the dataset. Many applications, particularly in a medical context, require models to generalise to new participants—e.g., a diagnostic model is of little use if it can only diagnose patients in the training data. Consequently, if there is a significant distribution shift between participants,

---

[1]`https://github.com/bomatter/participant-diversity-paper`

39th Conference on Neural Information Processing Systems (NeurIPS 2025).

low participant diversity[2] could impede model performance even with a large overall sample size. Understanding the relevance of participant diversity is critical to guide data collection and future machine learning work. If models generalise well across participants, data could be collected from smaller cohorts, significantly reducing costs and logistical challenges. Conversely, if participant diversity is crucial, this motivates not only higher participant counts during data collection, but also the development of machine learning strategies that explicitly address participant distribution shifts.

We address these challenges by systematically investigating the data scaling behaviour of different EEG machine learning models (TCN, mAtt, LaBraM) across a range of large datasets (TUAB, CAUEEG, PhysioNet), each with more than 1,000 participants, and three tasks (EEG normality prediction, dementia diagnosis, sleep staging). Crucially, we control both the number of participants in the training data and the amount of data per participant, allowing us to disentangle the impact of participant diversity from the overall sample size. Furthermore, we assess the utility of existing data augmentations (AmplitudeScaling, FrequencyShift, PhaseRandomisation) and a pre-trained foundation model (LaBraM) for varying amounts of (downstream) training data. Specifically, we quantify their effectiveness in data-limited regimes, whether they continue to provide value as the dataset size increases, and how they address specific limitations, such as low participant diversity versus limited data per participant.

In summary, we formalise the data generation process for EEG-based machine learning and use recent, relevant advances in statistical learning theory to inform our experiment designs and answer the following research questions:

- **Q1:** *Does a lack of participants in EEG datasets substantially limit the performance of machine learning models?*
- **Q2:** *Is collecting more data per participant a reliable way to compensate for a lack of participants?*
- **Q3:** *Can contemporary EEG data augmentations effectively increase performance across different data regimes, including limited participant counts or data per participant?*
- **Q4:** *Can self-supervised pre-training effectively increase performance across different data regimes, including limited participant counts or data per participant?*

Our findings provide useful insights to guide data collection and the development of EEG-based machine learning methods.

## 2   Related Work

**EEG Data Scaling Studies**   Banville et al. have studied scaling laws for image decoding from brain activity measured with EEG, MEG, or fMRI [7]. A critical difference to this work is that they trained participant-specific models, where data from the same participant is used during training and test time and models do not have to generalise to unseen participants. Our work is focused on settings where generalisation to new participants is crucial, such as applications like diagnosis or prognosis. Kiessner et al. specifically investigated the scaling behaviour of convolutional neural networks for EEG normality prediction on the TUAB dataset and presented scaling laws for model and data size [9]. While their experiments controlled for data size in terms of the number of recordings, we aim to disentangle the overall sample size, controlling for both the number of participants and the amount of data per participant. This allows us to study the relevance of participant distribution shifts and the need for sufficient participant diversity in datasets used for model training. Our conclusions are also more broadly applicable, as we study different tasks, datasets, and more diverse model architectures.

**EEG data augmentations**   Several works have proposed and employed EEG data augmentations [6, 10, 11, 12, 13, 14]. Mohsenvand et al. introduced several augmentations in the context of a contrastive learning approach for EEG [12]. Rommel et al. benchmarked various EEG data augmentations on a sleep staging dataset with 78 participants and a motor imagery task with 9 participants [10]. We build on their work, including two promising augmentations from Rommel et al. and one from Mohsenvand et al. in our experiments. Our work is different in that we specifically investigate the utility of data augmentations in dealing with limited participant counts and limited data per

---

[2]In this study, 'participant diversity' refers to the number of individuals contributing data to the training dataset, with higher diversity corresponding to a greater number of unique participants.

participant. Furthermore, our experiments were conducted on much larger datasets with over 1,000 participants each, allowing us to study the impact of augmentations across a broader range of data regimes. Finally, their experiments were limited to a single model per task, whereas we benchmarked all augmentations with different model architectures for each task and observed differential effects.

**Self-supervised pre-training**    Recently, a lot of work has also been published on pre-trained EEG (foundation) models [15, 16, 17, 18, 19], most of which is based on transformer models operating on patches that correspond to channel-wise EEG segments and masked token prediction. Our experiments include LaBraM as a state-of-the-art example of these approaches and provides an independent evaluation of the utility of self-supervised pre-training across different data regimes. Furthermore, our aim was again to disentangle the effect of participant counts vs overall sample size.

## 3   Methods

We model datasets used for EEG-based machine learning by the hierarchical data generation process illustrated in Figure 1A. A dataset comprises recordings from $n$ different participants, which are in turn segmented into $m$ distinct samples per participant. As a result, the overall sample size is usually substantially larger than the number of individual participants from which the samples are obtained. Formally, we can denote the sample features by $X_{ij}$ where $i = 1, ..., n$ indexes the participants, and $j = 1, ..., m$ indexes the samples for participant $i$, and with corresponding ground-truth labels denoted by $Y_{ij}$. For each participant, these samples follow a participant-specific distribution $P_i$, which is itself drawn from a population-level distribution $Q$. We use a loss function, $\ell(\hat{Y}, Y)$, to measure the quality of a prediction, $\hat{Y} = f(X)$, provided by some model, $f$. This could be, e.g., the zero-one loss or the cross entropy loss, and need not be the loss actually used to train a model. We can define the empirical risk (measured on the training set) and population risk, respectively, as

$$r(f) := \frac{1}{mn} \sum_{i=1}^{n} \sum_{j=1}^{m} \ell(f(X_{ij}), Y_{ij}) \quad \text{and} \quad R(f) = \mathbb{E}_{X,Y}[\ell(f(X), Y)]. \tag{1}$$

In the classic learning setting, where each $P_i$ is assumed to be the same, it is well established in the statistical learning theory literature that the gap between empirical risk and population risk (i.e., the extent to which a model will overfit) scales as $\tilde{\Theta}(1/(mn)^p)$, where $p \in \{\frac{1}{2}, 1\}$ [20]. The value of $p$ will be $\frac{1}{2}$ unless some assumptions are satisfied relating to the suitability of the model architecture and the amount of information the features contain about the labels [21]. The two-level data generation process we consider is comparatively less studied, but it is known that with minimal assumptions about model choices and feature information content, the gap will scale as $\tilde{\Theta}(1/n^{\frac{1}{2}} + 1/(mn)^{\frac{1}{2}})$ [22]. We include the second term, which is asymptotically dominated by the first, because if the distribution shift is small, or one is able to select a model architecture with substantial robustness to distribution shifts, then the second term can be larger in the non-asymptotic regime [22]. This analysis tells us that in the presence of a substantial distribution shift, the extent to which one should expect to overfit is governed primarily by the number of participants, rather than the amount of data available per participant, unless one is able to design a model architecture with the appropriate invariances. Under this framework, we address the four research questions outlined in the introduction. We assess the relevance and magnitude of participant distribution shifts through the scaling behaviour of model performance (Q1 & Q2) and the effectiveness of augmentations (Q3) and self-supervised pre-training (Q4) to improve performance by endowing models with useful invariances or boosting effective participant counts.

### 3.1   Experimental Design and Model Training

We conducted data scaling experiments to empirically assess the dependence of ML model performance on the participant count $n$ and the number of samples per participant $m$ across three large datasets and prediction tasks (see Section 3.2). Each dataset was split into train, validation, and test splits with no participant overlap. Subsequently, we randomly subsampled training datasets with a fixed number of participants and overall sample size from the train split as visualised by the grid depicted in Figure 1B. Different ML models (see Section 3.4) were then trained on the subsampled training data (see Appendix A.1 for details and hyperparameters). Early stopping (on the

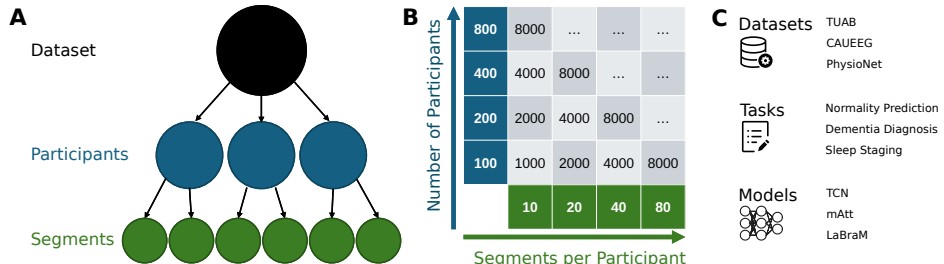

Figure 1: **(A)** Multi-level EEG data distribution. A dataset contains EEG recordings from multiple participants, which are divided into smaller segments to train machine learning models. **(B)** Grid visualising the subsampling of training datasets controlling for both the number of participants and the number of segments per participant. **(C)** The datasets, tasks, and models used in the experiments.

fixed, shared validation set) was used to prevent overfitting with limited amounts of training data, while still allowing training to converge for larger training datasets. To obtain uncertainty estimates, each trial—i.e. each combination of participant count and number of samples per participant—was repeated multiple times with different random seeds for the training data subsampling. Finally, all trained models were evaluated on the same test data. To further study different strategies to deal with limited data, we repeated the same procedure for each studied method designed to address a lack of data (data augmentation: see Section 3.5, self-supervised learning: see Section 3.6).

### 3.2 Data and Tasks

**TUAB**    The Temple University Hospital Abnormal (TUAB) EEG Corpus [23, 24], a subset of the TUH EEG Corpus [25], was collected in a hospital environment and is provided with labels about the normality of the EEG according to standardised criteria [26]. The full dataset contains more than 2300 participants with most recording durations in the range of 15-25 minutes. For the current study, we used version 3.0.1 of the dataset. Data access can be requested from the dataset authors.

**CAUEEG**    The Chung-Ang University Hospital EEG (CAUEEG) dataset [27] comprises data from 1379 participants with most recording durations in the range of 5 - 20 minutes. In the current study, we used a subset of data from 1122 participants for which diagnostic labels for the categories normal, mild cognitive impairment (MCI), or dementia were available. Diagnostic labels were assigned by neurologists based on a neuropsychological examination. Data access for academic and research purposes can be requested from the dataset authors.

**PhysioNet**    The dataset referred to as PhysioNet in this study corresponds to the data used in the PhysioNet/Computing in Cardiology Challenge 2018 [28, 29]. While the official challenge was focused on the classification of arousal regions, we focused on sleep staging here. The dataset comprises polysomnographic (PSG) recordings along with sleep stage annotations for non-overlapping 30-second epochs following AASM standards. While the PSG recordings contained additional measurements such as electrooculography (EOG), electromyography (EMG), or airflow and oxygen saturation, only the six EEG channels were used in this study. Furthermore, we only used the training data of version 1.0.0 of the dataset, as annotations for the official test split were not made publicly available. The data is available under the Open Data Commons Attribution License (ODC-By) v1.0.

### 3.3 Preprocessing

Preprocessing of the data comprised the following steps: Band-pass filtering to 1-50 Hz and resampling to a sampling frequency of 100 Hz, cropping (to 15 minutes for TUAB, omitting the first 30 seconds due to frequent strong artefacts at the beginning of recordings; to 5 minutes for CAUEEG), selection of a fixed channel subset (the 19 10-20 channels for TUAB and CAUEEG; the 6 EEG channels for PhysioNet), re-referencing to average reference, and segmentation into non-overlapping epochs (2 seconds for TUAB and CAUEEG; 30 seconds for PhysioNet).

For compatibility with the pre-trained LaBraM model, the following adjustments were made for the training and fine-tuning of LaBraM: amplitudes scaled to units of 0.1 mV, sampling frequency of 200 Hz, filtering to 0.1-75 Hz with a notch filter at 50 Hz (see [15]). We note that the notch filter was

not matched to the line noise of the data in the original publication (e.g. LaBraM was also evaluated on TUAB, where line noise is at 60 Hz). For compatibility with the pre-trained weights, we did not correct this potentially suboptimal setting. Furthermore, since LaBraM only supports segment lengths up to 16 seconds without modification (due to the implementation of the temporal positional encodings), we used 15 second epochs instead of 30 second epochs for PhysioNet.

## 3.4 Models

We selected three influential and widely used models with distinct architectures to obtain a diverse perspective on EEG model behaviour and reduce the risk of model-specific effects.

**TCN** The temporal convolutional network (TCN) architecture was originally proposed by Bai, Kolter, and Koltun [30] and has since been adapted and widely used for EEG data [1]. The model consists of a stack of 1D causal convolution layers. We used the braindecode implementation [31] with the following hyperparameters: n_blocks=4, n_filters=64, kernel_size=5, drop_prob=0.

**mAtt** The manifold attention network (mAtt) is a geometric deep learning model developed for EEG data [32]. Briefly, the architecture starts with two convolutional layers and then encodes segments in the input EEG into a sequence of covariance matrices. The diagonals in these covariance matrices correspond to the signal power in each channel. The architecture then makes use of the fact that the covariance matrices are symmetric positive definite (SPD) through a manifold attention module, where instead of dot product attention, an approximation of the Riemannian distance is used. The output is then transformed back to Euclidean space and passed through a fully connected layer to get classifications. We adapted the code from the official mAtt repository (https://github.com/CECNL/MAtt) and used 100 kernels in conv1, 50 kernels in conv2 with a kernel size of 11, and an embedding size of 25 for the SPD transforms.

**LaBraM** The Large Brain Model (LaBraM) architecture was designed by Jiang, Zhao, and Lu [15]. The model operates on patches corresponding to channel-wise time segments. Patches are first passed through a small convolutional encoder and spatial and temporal positional encodings are added. The resulting tokens are then processed through a stack of transformer encoder layers and average pooling is used to aggregate the output tokens, followed by a classification head. We used the LaBraM Base model with the recommended default hyperparameters.

## 3.5 Augmentations

We selected three promising augmentations based on prior work (see Appendix A.2 for details).

**Amplitude Scaling** The AmplitudeScaling data augmentation scales (multiplies) each EEG channel by a random factor $a \in [0.5, \ 2]$. This augmentation was proposed by Mohsenvand, Izadi, and Maes and used for contrastive learning [12]. They consulted four neurologists to identify augmentations that would not change the interpretation of the data along with recommended ranges for the parameters (min and max scaling factor in this case; we used the same settings in our experiments).

**Frequency Shift** FrequencyShift was introduced in [10] and further evaluated in [6]. This augmentation introduces a small shift in the frequency domain, i.e. the power spectrum is shifted by some random frequency $\Delta f$. The same shift is applied to all EEG channels. Rommel et al. tuned the hyperparameter defining the maximum shift for a sleep staging task and achieved the best results with $\Delta f_{max} = 0.3$ Hz. We used the same parameter in our experiments.

**Phase Randomisation** Schwabedal et al. introduced the randomisation of Fourier transform phases for data augmentation under the name of "FT surrogates" [13]. We refer to this augmentation as PhaseRandomisation. Rommel et al. evaluated different hyperparameters for the maximum phase shift and observed the best results with high values [6]. We adopt the maximum range ($\Delta \varphi_{max} = 2\pi$), which corresponds to complete randomisation of the phase information. We also sample different phases for each EEG channel, the setting that Rommel et al. used for their sleep staging experiments.

## 3.6 Pre-training

The LaBraM model (see Section 3.4) was developed as a foundation model for EEG data and pre-trained to predict discrete tokens, previously learned by a separate tokeniser that was itself trained to predict the Fourier spectrum (see the original publication for details [15]). The model was pre-trained

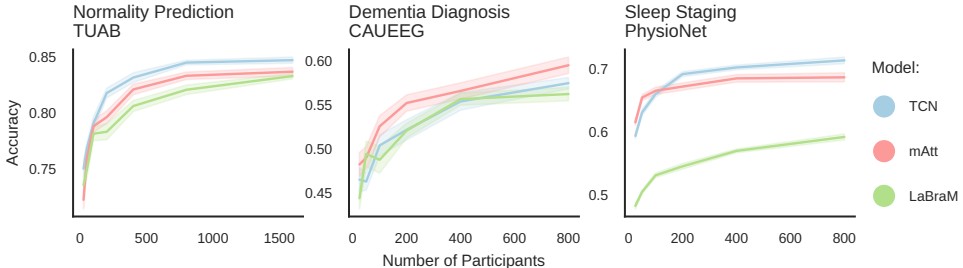

Figure 2: **Scaling behaviour of model performance.** Average accuracies of the different models for normality prediction on TUAB, dementia diagnosis on CAUEEG, and sleep staging on PhysioNet for increasing participant counts at a fixed number of segments per participant ($m = 40$). Averages were computed across seeds used to subsample the training datasets and the shading illustrates the standard error of the mean. Across all datasets, performance increased strongly as the size was increased to several hundred participants, after which improvements started to diminish.

in an unsupervised manner on a collection of 16 datasets with a total of around 500 participants, and the weights for the smallest version (LaBraM Base) were made publicly available by the authors. We used these weights for experiments with the pre-trained LaBraM model.

The model was then fine-tuned using the labelled training data of the corresponding task. We fine-tuned the entire architecture without freezing any layers and determined the learning rate through hyperparameter optimisation. Specifically, we evaluated a grid of learning rates [1e-7, 5e-7, 1e-6, ..., 1e-2] for each dataset (TUAB, CAUEEG, PhysioNet) using the respective training and validation splits. To ensure that performance differences between the pre-trained and baseline (trained from scratch) models can be attributed to pre-training rather than tuning of the learning rate, we also tuned the learning rate for the baseline in the same way. Table 1 shows the selected learning rates.

## 4 Results

### 4.1 Scaling the Number of Participants

**Q1:** *Does a lack of participants in EEG datasets substantially limit the performance of machine learning models?*

Figure 2 depicts the model performance as the number of participants in the training data is varied while keeping the number of segments per participant fixed. We observe strong improvements in accuracy as the participant count increased from 25 to several hundred participants across all datasets and models, after which accuracy improvements diminish. While performance appears not to have fully saturated across the three tasks, meaningful further improvements through the collection of additional labelled training data from new participants is likely infeasible. For example, recruiting an additional 1,600 participants to double the number of participants in the TUAB experiments would often be prohibitively expensive for an expected improvement in performance of roughly 1%. To put this into context, Roy et al. reported the sizes of datasets used across 154 studies on deep learning-based EEG analysis [5]. Astonishingly, half of the datasets contained data from less than 13 participants and only six comprised more than 250 participants. Although this might have changed since the review's publication in 2019, according to our experimental results, a substantial proportion of EEG datasets fall into the range where a scarcity of participants is likely a critical limiting factor. In such cases, substantial gains could be achieved through the collection of additional training data from more participants. Differences between model architectures are discussed in Appendix A.5. Overall, accuracies were relatively similar across architectures and the sample size was the primary driver of performance.

**A1:** *Yes. Across all datasets, tasks, and models, performance improved substantially as the participant count was increased to several hundred participants. The point where performance began to saturate was substantially higher than the number of participants seen in many EEG datasets, indicating that collecting data from more participants could lead to non-trivial gains in performance in many typical contexts.*

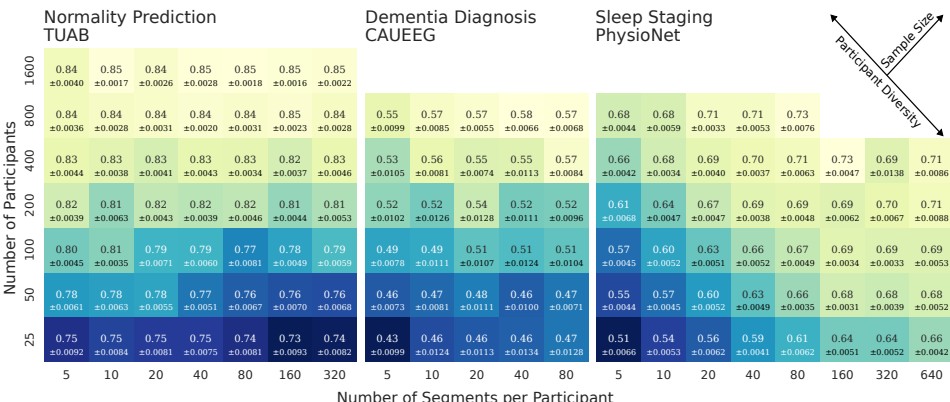

Figure 3: **Differential effects of participant count and overall sample size on model performance.** Accuracies of the TCN model for normality prediction on TUAB, dementia diagnosis on CAUEEG, and sleep staging on PhysioNet averaged across seeds, along with the standard error of the mean. Performance on TUAB and CAUEEG was dominated by the participant count. Sleep staging performance on PhysioNet was more dependent on the overall sample size and competitive performance was achieved even with very limited participant counts.

## 4.2 Scaling Segments per Participant

**Q2:** *Is collecting more data per participant a reliable way to compensate for a lack of participants?*

Figure 3 illustrates the performance of the TCN model when varying both $m$ and $n$. Additional results using the mAtt and LaBraM models are provided in Appendix A.8, where we observe qualitatively similar trends. Focusing first on normality prediction on TUAB, we observe that increasing the amount of data for a fixed participant count (moving along the horizontal axis) did not yield any substantial performance improvements. As already observed in Section 4.1, performance was highly dependent on the participant count (moving along the vertical axis). Even for a fixed sample size, higher participant diversity yielded significant improvements (moving diagonally from bottom right to top left). The results for the CAUEEG dementia prediction task exhibit a similar pattern. Linking back to the theoretical analysis in Section 3.1, this suggests a substantial distribution shift between participants such that the participant count $n$ dominated model generalisation.

We observe a different trend for PhysioNet sleep staging: higher participant diversity still yielded some improvements in performance (moving diagonally from bottom right to top left), but substantial improvements were also achieved by increasing the sample size for a fixed participant count (moving along the horizontal axis). In fact, with the maximum number of samples per participant, training data from a small cohort of just 25 participants was sufficient to match the performance achieved with training data from 400 participants (with 5 segments per participant).

**A2:** *No. In two of the three case studies that we investigated, scaling the amount of data per participant had minimal impact on the performance of the model. The number of participants in the training datasets had a substantially larger impact on model performance.*

## 4.3 Effectiveness of EEG Data Augmentation

**Q3:** *Can contemporary EEG data augmentations effectively increase performance across different data regimes, including limited participant counts or data per participant?*

Figure 4 illustrates the effect of different data augmentations (see Section 3.5) on the resulting model performance. With one exception, the impact of the different augmentations was small and inconsistent. No significant associations between sample size and accuracy improvements were observed (see Appendix A.9). What stood out, however, was a systematically detrimental effect of the PhaseRandomisation augmentation on PhysioNet for the TCN and mAtt models, whereas performance for the LaBraM model was improved. LaBraM also benefitted from the FrequencyShift augmentation on PhysioNet. In terms of absolute performance, LaBraM performed substantially worse on PhysioNet without data augmentation (see Figure 2), such that with data augmentation,

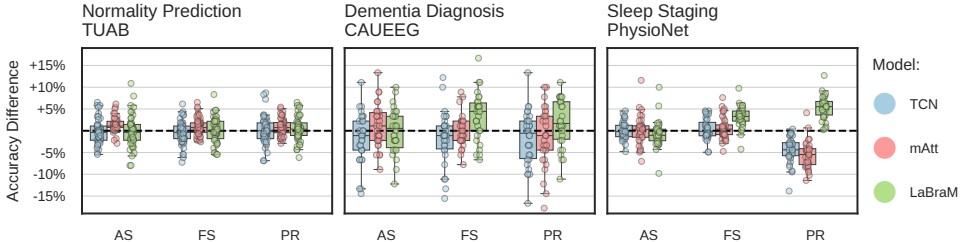

Figure 4: **Impact of data augmentations.** Each boxplot shows pairwise accuracy differences between augmented and unaugmented training (positive means augmentation improved performance) across all combinations of participant counts ($n$) and segments per participant ($m$). Results for a single seed are shown for better visibility. AS = AmplitudeScaling, FS = FrequencyShift, PR = PhaseRandomisation. On PhysioNet, the LaBraM model benefitted from FS and PR, whereas PR decreased performance for TCN and mAtt. Other than that, augmentations did not lead to consistent improvements in performance.

all three model architectures perform about equally well. The observed performance improvements for LaBraM can therefore likely be attributed to its greater reliance on large amounts of data, which is partially offset by augmentation. Decreased performance in the smaller and more data-efficient models suggests that PhaseRandomisation destroys useful information.

The PhaseRandomisation augmentation randomises the phase of each Fourier coefficient and is applied channel-wise. It therefore preserves the power distribution across different frequencies, but completely destroys wave morphologies and cross-channel correlations. Given that certain waveforms are highly informative of the sleep stage (e.g. K-complexes are known to occur during N2 sleep), the performance decreases should be expected in such situations.

A likely explanation for why AmplitudeScaling and FrequencyShift did not yield consistent accuracy improvements is that these augmentations are not truly label preserving. For example, overall amplitude and differences in amplitude between EEG channels (e.g. asymmetries between corresponding channels on the left and right hemisphere) can provide clinically relevant information [26]. Similarly, frequency shifts can be clinically relevant (e.g. slowing may reflect abnormalities [26] and has been linked to MCI and AD [33, 34]). Alternatively, it is possible that even the smallest amount of training data already provided sufficient diversity of amplitudes and frequency shifts for the model to learn sufficient invariances to these aspects of the data, rendering the augmentations redundant.

**A3:** *No. The augmentations benchmarked in this study failed to consistently improve performance. Small gains were observed for the largest model on a single task, where it still underperformed compared to the smaller models trained without augmentations.*

### 4.4 Effectiveness of Self-supervised Pre-training

**Q4:** *Can self-supervised pre-training effectively increase performance across different data regimes, including limited participant counts or data per participant?*

Figure 5 shows a comparison of the LaBraM model performance with and without pre-training (see Section 3.6) on the TUAB normality prediction task. Pre-training consistently improved model performance across all data regimes, except when the amount of downstream fine-tuning data was most severely limited. Comparing the baseline trained from scratch models (left heat map) and the pre-trained models (middle heat map), the effect of pre-training looks like a boost in the amount of data per participant: with only 10-20 segments per participant, the pre-trained model achieved the same performance as the baseline model with the maximum amount of 320 segments per participant. However, pre-training also improved performance when the participant count was low and it continued to provide value even when the maximum amount of data was used. Consistent results were observed for dementia prediction on the CAUEEG dataset and sleep staging on PhysioNet (see Appendix A.10).

In Figure 2, we observed that the LaBraM model required substantially more data to achieve the same performance as the mAtt and TCN models. With pre-training, however, the much larger and more data-demanding LaBraM model was able to match or even outperform them. This can be seen by comparing the heatmaps in Figure 5 to the corresponding heatmap on TUAB of the TCN model

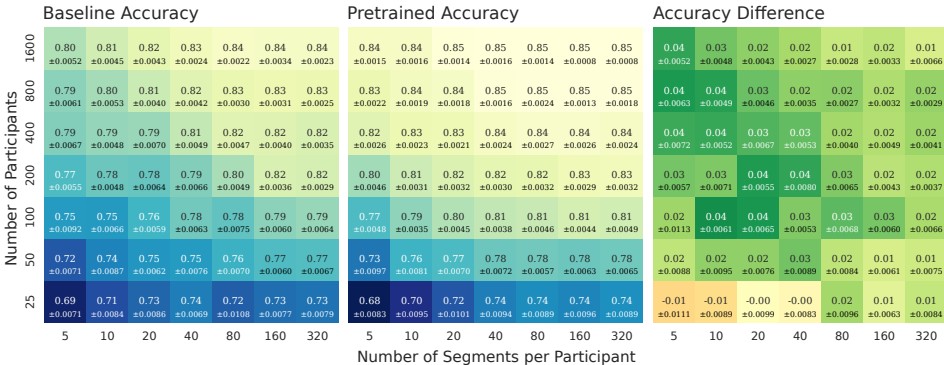

Figure 5: **Effectiveness of self-supervised pre-training.** Comparison of LaBraM performance with and without pre-training on TUAB. The left and middle heatmaps show average accuracies (± standard error of the mean) for LaBraM trained from scratch (baseline) and pre-trained on a collection of 16 datasets before fine-tuning on TUAB, respectively. The right heatmap visualises the average of pairwise accuracy differences (± standard error of the mean), where pairs correspond to the accuracies of pre-trained and baseline models for the same seed. Pre-training consistently improved performance across all data regimes, except when the amount of fine-tuning data was most severely limited.

in Figure 3. Alternatively, Appendix A.6 shows a version of Figure 2 with additional lines for the pre-trained LaBraM model.

Pre-training exposed the model to a larger diversity of EEG data from additional participants than those seen during fine-tuning. The positive results suggest that, even without labels, this has allowed the model to learn useful representations that transfer to diverse downstream tasks.

**A4:** *Yes. Self-supervised pre-training led to consistent improvements in model performance across all data regimes except when fine-tuning data was most severely limited.*

## 5 Discussion

The present study demonstrates that participant diversity is a critical factor for training EEG-based machine learning models. In two out of three tasks studied, collecting more data per participant was not effective to compensate for a small participant count. At the same time, our findings also show that good performance can be achieved with limited participant cohorts in specific applications. Pilot studies can help identifying such cases and inform the extent to which participant diversity should be prioritised in data collection and methodological machine learning work.

The use of large datasets allowed us to study the scaling behaviour of machine learning models over a large range of data regimes. Across all datasets, tasks, and models, we observed strong improvements in performance as the participant count was increased from 25 to several hundred participants. This highlights the value of initiatives like AI-Mind [35], a collaboration of 15 academic and industry partners pioneering a large study involving 1,000 participants with mild cognitive impairment. However, as participant counts approached the maximum sizes explored in our study, performance improvements began to diminish. While the majority of current EEG-based machine learning work still operates on smaller datasets, it is thus possible that further performance improvements through the collection of even larger labelled datasets will become prohibitively expensive. Complementary approaches, such as machine learning strategies explicitly designed to address participant distribution shifts and make more effective use of limited EEG data could become essential.

Based on these findings, we assessed the effectiveness of existing machine learning strategies for dealing with limited participant diversity and data per individual participant. In contrast to other areas where ML is used, data augmentations did not prove effective as a strategy to deal with limited participant diversity or overall sample size. It is important to note that our experimental setup considered only existing EEG data augmentation methods; we do not exclude the possibility that more effective augmentations can be designed. In fact, our results encourage the development of augmentations that specifically imitate participant differences.

Self-supervised pre-training consistently improved model performance across all data regimes. These results are particularly promising considering the broad potential for further improvements: This study benchmarked LaBraM Base (the smallest version of the model), for which weights were made publicly available by the authors. Further improvements are thus likely achievable with increased model size as indicated by comparisons of the model versions in the original publication [15] as well as generally observed trends in self-supervised learning [36]. Moreover, the data used to pre-train LaBraM is still relatively limited with a total participant count of around 500—less than a third of the maximum number of participants used in our evaluations on the TUAB dataset. The existing body of literature on self-supervised learning strongly suggests that more data would result in better performance [36]. Since pre-training does not require labelled data, the creation of a larger pre-training dataset is comparatively easy. Prior work has also shown that self-supervised learning can be used to enforce approximate invariances [37], which can help circumvent the poor dependence of overfitting behaviour on the number of participants in the training data [22].

## 6 Limitations and Future Work

While our study provides insights into the scaling behaviour of EEG models, the importance of participant distribution shifts, and the utility of data augmentations and self-supervised learning to address EEG data limitations, several limitations remain that warrant further investigation.

First, our study focuses on applications of EEG that require models to generalise to unseen participants. We see this as a particularly important setting, as it is essential for most clinical applications such as diagnosis or prognosis. However, there are other domains—especially brain-computer interface (BCI) applications—where it is common and often practical to train participant-specific models. In concurrent work, Banville et al. have studied scaling laws for image decoding from brain activity and observed little gain from increasing the number of participants in the training data, although their experiments were limited to three datasets with only 8-48 participants [7]. Kong et al. have observed more promising generalisation to unseen participants with larger datasets [38].

Beyond the augmentations and self-supervised learning methods studied in our work, refined preprocessing methods could provide another approach to dealing with limited EEG data by improving data quality instead of quantity. There is a rich literature on EEG preprocessing [39, 40, 41], although most methods were designed for classical EEG analysis and it is unclear how well they translate to ML-based EEG applications. Some work suggests that even with large datasets, machine learning models still benefit in particular from the removal of high-amplitude artefacts [42, 43].

Finally, future research should attempt to determine the nature of participant distribution shifts. We observed a large participant count to be critical for model performance in EEG normality prediction and EEG-based dementia diagnosis, and we can see several plausible underlying reasons. On the one hand, anatomical individual differences could impede the generalisability of the observed patterns of scalp activity measured with EEG. On the other hand, the distribution shift might also be explained by variability in how the target presents in different participants. For instance, in the EEG normality prediction task, there are numerous reasons why an EEG can be abnormal, ranging from focal epileptiform discharges to diffuse slowing. A model that is trained on data from only a small cohort of participants will therefore not only be exposed to little demographic and anatomical variation but also to a limited number of target (or disease) phenotypes. This could be particularly relevant, e.g., in the context of neurodegenerative or psychiatric diseases, where substantial individual differences in symptomatology and neural correlates have been described [44, 45].

## 7 Conclusion

Our study shows that participant diversity is a key determinant of EEG-based machine learning performance, with task-dependent importance. While increasing data per participant cannot compensate for low participant counts in most cases, pilot studies can help to identify contexts where smaller cohorts suffice. Furthermore, while performance generally scales with participant numbers, gains diminish at larger scales, suggesting limits to dataset growth as a strategy and motivating more work on machine learning methods to deal with limited EEG data. Particularly self-supervised pre-training provided consistent benefits in our experiments, highlighting its promise alongside approaches designed to explicitly address participant distribution shifts.

## Acknowledgements

This project was supported by the Royal Academy of Engineering under the Research Fellowship programme.

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

# A    Technical Appendices and Supplementary Material

## A.1    Training Details and Hyperparameters

The official train-test splits were used for TUAB and CAUEEG, and in both cases, part of the train split was set aside for validation. For PhysioNet, where labels for the official test split are not publicly available, an age and sex stratified split into train, val, and test was created on the participant level (i.e. without overlapping participants). The exact splits can be reproduced with the shared research code.

Models were trained using AdamW (`betas=(0.9, 0.999)`, `weight_decay=0.01`), global gradient norm clipping (`max_norm = 1.0`), and a cross-entropy loss. The learning rate was set to `1e-3` for mAtt and TCN and tuned for LaBraM (see Section 3.6 and Table 1 for details and the used learning rates respectively). For the CAUEEG dataset, we used class weights in the loss function to account for class imbalance. Since the amount and diversity of the training data differed substantially between trials, early stopping (on the fixed validation set) was used to prevent overfitting, while allowing sufficient training time if the amount of data was larger. Specifically, the validation loss was evaluated every 500 batches and a patience of 5 was used to trigger early stopping.

Table 1: Learning rates for the LaBraM model.

| Dataset | From Scratch | Fine-tuning |
|---|---|---|
| TUAB | $1e$-4 | $1e$-6 |
| CAUEEG | $1e$-3 | $5e$-6 |
| PHYSIONET | $1e$-3 | $1e$-5 |

All models were trained on an internal cluster with RTX 2080 Ti (11GB) and A40 (48GB) GPUs. Training times varied given the different models, dataset sizes, and the use of early stopping. We estimate the total runtime to reproduce the baseline results at around a week on 20 RTX 2080 Ti GPUs and at around two weeks for all results, including the augmentation, pre-training, and ablation experiments.

## A.2    Selection of EEG Data Augmentations

The selection of EEG data augmentations investigated in our work was informed by prior work. PhaseRandomisation (termed FTSurrogate) consistently emerged as the most effective augmentation in the systematic comparison on two tasks (sleep staging and BCI motor imagery) by Rommel et al. [6]. FrequencyShift showed more modest gains, but is particularly interesting to study in the context of participant diversity, because differences in peak frequencies are associated with demographic variables like age and sex [46, 47, 48]. Importantly, other augmentations like GaussianNoise, ChannelSymmetry, ChannelShuffle, TimeMasking, and BandstopFilter did not show consistent and significant improvements in their comparison, which is why we did not include these augmentations in the present work. The two augmentations SignFlip and TimeReverse showed some improvements in limited settings, but were not included in our study because they clearly violate the structural integrity of EEG data. Furthermore, while ChannelDropout seemed to provide limited value in the BCI task, a closer look revealed that the dropout rate was tuned to 1 (i.e. dropout of all channels), hinting at an issue with the evaluation task or the training, rather than genuine augmentation benefit. Finally, we also studied AmplitudeScaling [12], which was originally proposed in the context of a contrastive learning approach and was not part of the benchmarks by Rommel et al.

## A.3    Metrics

We assessed model performance through prediction accuracy. For TUAB and CAUEEG, segment predictions were first aggregated into participant-level predictions via majority voting, such that the average was then computed across participants rather than across all pooled segments. We chose this approach because the labels were originally assigned on the participant level rather than to individual segments within the recordings, and because it better captures a model's ability to diagnose new participants.

As outlined in Section 3.1, each trial (i.e. each combination of a given number of participants and number of segments per participant) was repeated multiple times with different random seeds for the subsampling of the training data. This allowed us to report average accuracies along with the standard error of the mean across seeds. The baseline experiments were repeated with 25 seeds and the pre-training experiments were repeated with 25 seeds on TUAB and CAUEEG and with 5 seeds on PhysioNet. For the augmentations, we report pairwise comparisons for a single seed. On the PhysioNet dataset, where considerably longer recordings were available for some participants, we assessed conditions with up to 640 segments per participant. However, since this number of segments is not available for all participants in the dataset, experiments failed for some of the seeds, which is also the reason for the missing tiles in the top right corner of the PhysioNet results heatmaps. Rather than only reporting the results up to 40 segments per participant, up to which point we have complete results for all seeds, we included trials with failed seeds as they provide interesting and unbiased additional information.

Accuracy differences were computed as pairwise differences between corresponding results (same number of participants, number of segments per participant, and seed) across conditions (pre-trained vs baseline or augmented vs unaugmented). For average accuracy differences, these pairwise differences were averaged over seeds.

## A.4 Software

The research code for our experiments is publicly available at `https://github.com/bomatter/participant-diversity-paper`. Dependencies notably include PyTorch [49], MNE-Python [50] and MNE-BIDS [51] for data harmonisation and preprocessing, TorchEEG [52] for the implementation of the data loader, Braindecode [31] for the implementation of the TCN model and data augmentations, and the repositories of the original LaBraM [15] and mAtt [32] models.

## A.5 Model Comparison

One aspect determining the sample efficiency of a model is its architecture. Comparing model architectures in Figure 2, we observe that the TCN and mAtt architectures performed slightly better than LaBraM on TUAB and CAUEEG and markedly better on PhysioNet. On PhysioNet, LaBraM's substantially lower performance can partially be attributed to differences in preprocessing. As explained in Section 3.3, 15 second epochs had to be used for LaBraM instead of the 30 second epochs used for TCN and mAtt. We performed an ablation study where the TCN model was trained using the same preprocessing, which showed that differences in preprocessing explained part of the performance gap, but that TCN still outperformed the LaBraM model (see Appendix A.6). We hypothesise that LaBraM's lower performance stems from the weaker inductive bias of the transformer architecture and the fact that the LaBraM model is considerably larger in terms of the number of model parameters compared to the other models. The LaBraM model would thus be expected to be less sample efficient.

The differences between mAtt and TCN could reflect the usefulness of different priors for different tasks. TCN, as a convolutional neural network, is likely better at picking up on specific wave morphologies (e.g. sharp waves or K-complexes), which may be useful for normality prediction and sleep staging. The mAtt model, on the other hand, embeds the EEG into a sequence of covariance matrices that reflect signal power and cross-channel correlations, and might excel in tasks where these are the most relevant features.

On the whole, and with the exception of LaBraM's performance on PhysioNet, the achieved accuracies were relatively similar, despite substantial differences in the architectures and model sizes. The sample size was the primary driver of performance.

## A.6 Ablation: LaBraM-compatible Preprocessing

Data for the training and fine-tuning of the LaBraM model was preprocessed separately to ensure compatibility with the official pre-trained weights. On the PhysioNet dataset, this also meant that the segment length had to be reduced from 30s to 15s. Since LaBraM's performance on PhysioNet was particularly low compared to the other models, we performed an ablation study to see how much of the performance gap should be attributed to the different preprocessing rather than to architectural differences. Specifically, we trained the TCN model using the same preprocessing that was used for the LaBraM model. Figure 6 shows that performance decreased (light blue line vs dark blue line) but was still higher than the LaBraM baseline performance (light green line). With pre-training, LaBraM achieved comparable performance to the TCN model when the same preprocessing was used (dark blue and dark green lines). Overall, this suggests that the different preprocessing explains part of the performance gap, but not all of it. When models were trained from scratch, TCN still achieved better performance.

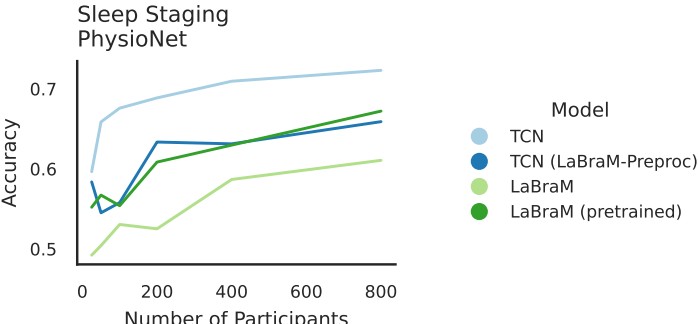

Figure 6: **Ablation: LaBraM-compatible preprocessing.** The light blue and dark blue lines represent TCN performance with standard and LaBraM-compatible preprocessing, respectively, while the light green and dark green lines show LaBraM performance without and with pre-training. The performance drop observed for TCN with LaBraM-compatible preprocessing suggests that preprocessing differences contributed to LaBraM's lower baseline performance. However, the fact that TCN still outperformed LaBraM when both used the same preprocessing indicates that architectural differences also played a role. Pre-training LaBraM mitigated the performance gap, allowing it to achieve results comparable to the TCN model when both used the same preprocessing.

## A.7 Control Experiment: Contiguous Segment Subsampling

In our main experiments, segments were subsampled uniformly at random. However, in practice, collecting fewer segments would usually correspond to shorter recordings, which are less likely to capture a wide range of long-term non-stationarities such as drifts, participant fatigue, or changes in mental state. Uniform subsampling may therefore artificially increase segment diversity compared to realistic data collection.

To examine whether this affects our conclusions, we performed a control experiment using contiguous subsampling, where training segments are drawn from consecutive blocks rather than uniformly. Specifically, for an epoched EEG with $N$ non-overlapping fixed-length segments and a desired number of segments $m$, we uniformly sample a random starting index $s \in [0, N - m]$. We then extract a contiguous block of $m$ segments: segments$[s : s + m]$.

Results are shown in Figure 7, which compares performance with uniform versus contiguous subsampling for normality prediction on TUAB with the TCN model. Patterns are consistent across both conditions, indicating that the conclusions are not affected by the choice of the subsampling strategy.

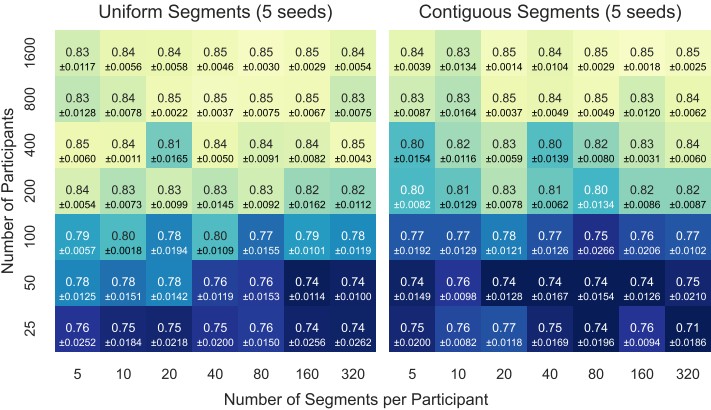

Figure 7: **Uniform versus contiguous segment subsampling.** Normality prediction accuracy on TUAB with the TCN model, averaged over 5 random seeds along with the standard error of the mean.

## A.8 Accuracy Heatmaps for mAtt and LaBraM

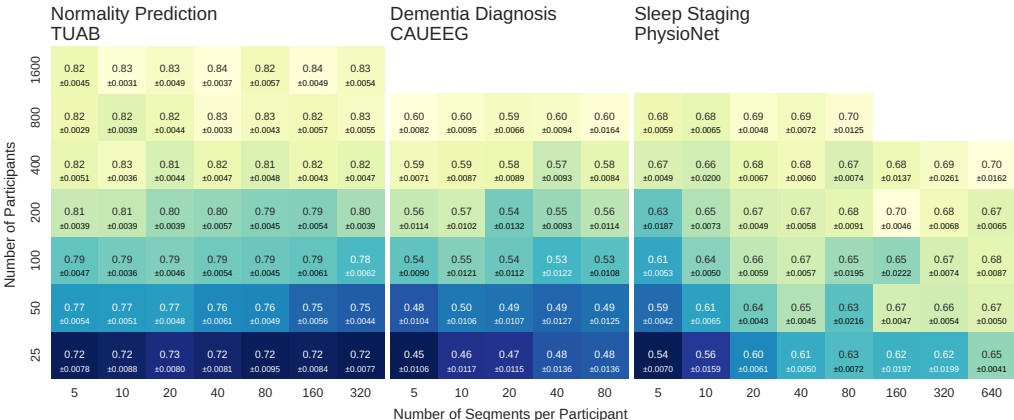

Figure 8: **Baseline performance heatmap for the mAtt model.** Accuracies for normality prediction on TUAB, dementia diagnosis on CAUEEG, and sleep staging on PhysioNet averaged across seeds, along with the standard error of the mean.

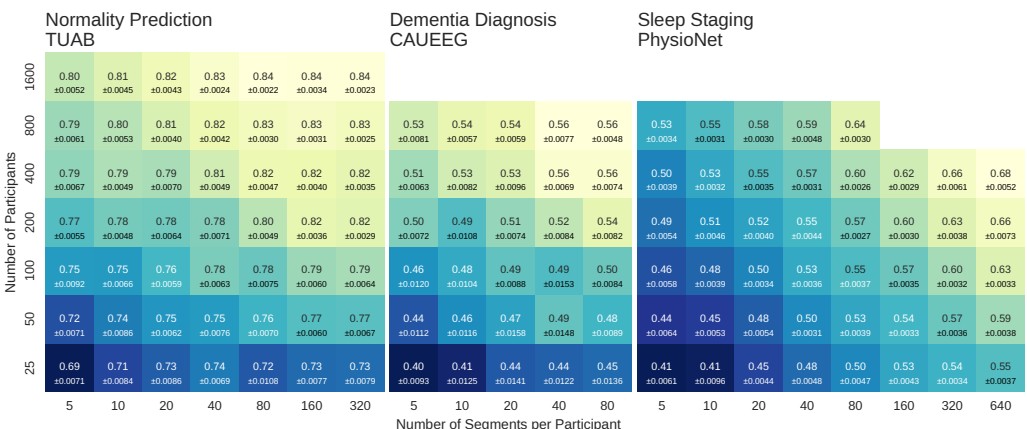

Figure 9: **Baseline performance heatmap for the LaBraM model.** Accuracies for normality prediction on TUAB, dementia diagnosis on CAUEEG, and sleep staging on PhysioNet averaged across seeds, along with the standard error of the mean.

## A.9 No Association between Sample Size and Accuracy Improvements with Augmentations

Table 2: Spearman correlations between sample size and accuracy difference to baseline performance with different augmentations. No associations between accuracy improvements and sample size were observed. $p$-values $< 0.05$ are marked with an asterisk. Note that these are uncorrected and none would remain significant under Bonferroni correction.

| Dataset | Augmentation | TCN | mAtt | LaBraM |
|---------|--------------|-----|------|--------|
| TUAB | AmplitudeScaling | 0.047 (p=0.746) | 0.079 (p=0.590) | -0.141 (p=0.335) |
| | FrequencyShift | 0.053 (p=0.717) | 0.174 (p=0.232) | 0.049 (p=0.738) |
| | PhaseRandomisation | 0.083 (p=0.571) | -0.076 (p=0.609) | 0.045 (p=0.761) |
| CAUEEG | AmplitudeScaling | -0.311 (p=0.095) | -0.195 (p=0.302) | 0.128 (p=0.499) |
| | FrequencyShift | -0.430* (p=0.018) | 0.055 (p=0.772) | 0.027 (p=0.885) |
| | PhaseRandomisation | -0.407* (p=0.026) | -0.324 (p=0.081) | -0.215 (p=0.253) |
| PhysioNet | AmplitudeScaling | 0.245 (p=0.155) | -0.339* (p=0.046) | -0.024 (p=0.886) |
| | FrequencyShift | -0.076 (p=0.663) | -0.037 (p=0.833) | 0.289 (p=0.075) |
| | PhaseRandomisation | -0.322 (p=0.059) | -0.144 (p=0.409) | 0.117 (p=0.503) |

## A.10 Pre-training Results on CAUEEG and PhysioNet

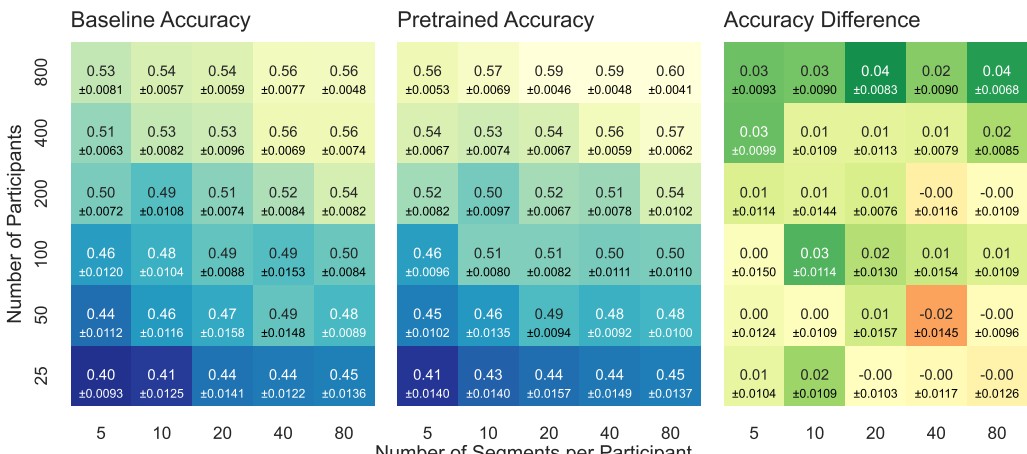

Figure 10: **Effect of pre-training on the CAUEEG dataset.** Comparison of LaBraM performance with and without pre-training on CAUEEG. The left and middle heatmaps show average accuracies (± standard error of the mean) for LaBraM trained from scratch (baseline) and pre-trained on a collection of 16 datasets before fine-tuning on CAUEEG, respectively. The right heatmap visualises the average of pairwise accuracy differences (± standard error of the mean), where pairs correspond to the accuracies of pre-trained and baseline models for the same seed.

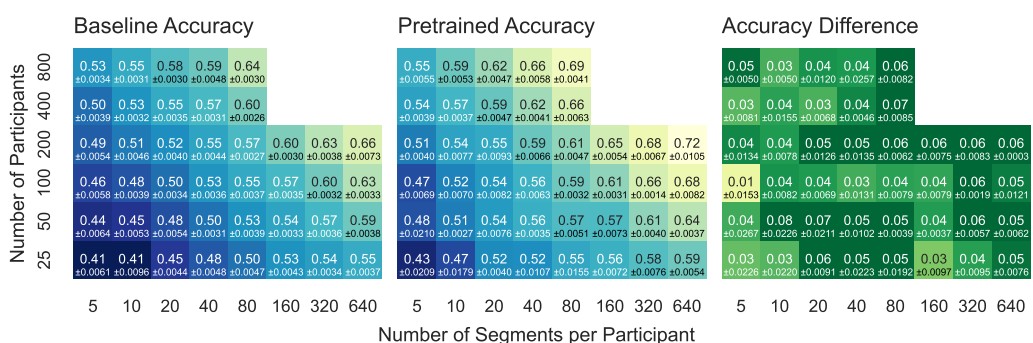

Figure 11: **Effect of pre-training on the PhysioNet dataset.** Comparison of LaBraM performance with and without pre-training on PhysioNet. The left and middle heatmaps show average accuracies (± standard error of the mean) for LaBraM trained from scratch (baseline) and pre-trained on a collection of 16 datasets before fine-tuning on PhysioNet, respectively. The right heatmap visualises the average of pairwise accuracy differences (± standard error of the mean), where pairs correspond to the accuracies of pre-trained and baseline models for the same seed.

## A.11    Scaling Behaviour Plots with Pre-trained LaBraM Model

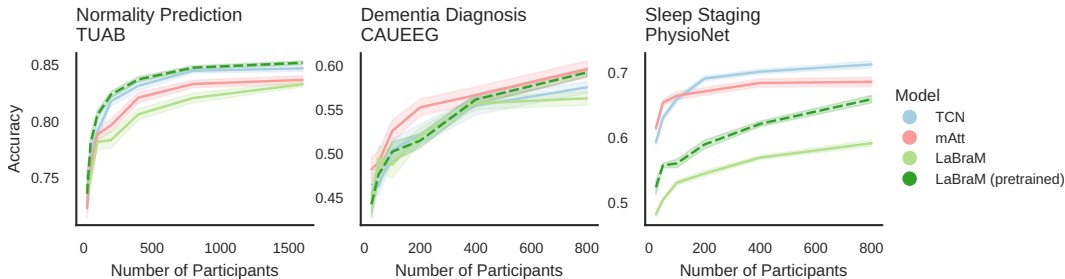

Figure 12: **Scaling behaviour with pre-trained LaBraM model.** Lines show the average accuracy across seeds with the standard error of the mean. Pre-training consistently improved LaBraM's performance and allowed it to achieve comparable or even superior results to the other models. The lower performance on the PhysioNet dataset is due to differences in preprocessing (see Appendix A.6).

