# OpenReview forum: "Is Limited Participant Diversity Impeding EEG-based Machine Learning?"
_NeurIPS.cc/2025/Conference — NeurIPS 2025 poster_

### Official Review · Reviewer_dTAg · 2025-06-06

**Clarity:** 3
**Significance:** 3
**Originality:** 2
**Rating:** 4
**Confidence:** 4

**Summary:**

The paper investigates how participant diversity impacts the performance and generalisation of machine learning models trained on EEG data. It frames EEG datasets as multi-level data structures, where large numbers of samples are derived from a small number of participants via windowing. The authors conduct large-scale empirical studies to disentangle the effects of total sample size versus number of participants. They demonstrate that increased participant diversity consistently leads to more robust generalisation, even when total sample size is held constant. The paper also evaluates common ML strategies like pretraining, domain adaptation and model architecture choices under varying diversity conditions to show that some methods are disproportionately sensitive to participant homogeneity.

**Questions:**

How is your analysis of augmentations under limited participant counts fundamentally different from standard augmentation studies aimed at data scarcity?

On what basis do you consider LaBraM indicative of broader self-supervised learning approaches in EEG?

Provide the derivation or source for the "~1% improvement" figure in line 239; is this drawn from your own results, external literature or an estimation?

Explain the rationale behind the choice of augmentation techniques. Why these four and what criteria were used to select them over others?

Clarify the scope of your conclusions on self-supervised learning. Are your findings intended to generalise beyond LaBraM and if so, how is that generalisation justified?

**Ethical Concerns:**

["NO or VERY MINOR ethics concerns only"]

**Final Justification:**

The authors have addressed a lot of the concerns addressed by the reviewers in their rebuttals.

**Limitations:**

The authors do include a limitations section but it is in the appendix and not integrated into the main discussion. The listed limitations are reasonable. They acknowledge the limited number of datasets and models used and the difficulty of decoupling all confounding factors in EEG data. However, these caveats are not reflected in the framing of their main conclusions, which sometimes overstate generality (e.g., claims about augmentation or self-supervised learning). These limitations should be surfaced earlier and more clearly tied to the scope of their findings.

**Quality:**

3

**Strengths And Weaknesses:**

Strengths:

- The paper systematically varies total data volume and number of unique participants to allow for controlled experiments that isolate the effects of diversity on model generalisation. This is very useful and not common in EEG literature.

- The paper assesses pretraining, domain adaptation, self-supervised learning and architecture variants across multiple datasets with a well-rounded perspective on how methods scale with diversity.

- By explicitly thinking of EEG datasets as nested hierarchies (participants -> sessions -> segments), the paper gives a helpful abstraction that clarifies why naïve sample count comparisons can be misleading.

Weaknesses:

- The authors claim in line 83 that their work is distinct because it investigates the utility of data augmentations under limited participant counts and limited data per participant. However, this distinction is unclear and weakly justified. Most augmentation methods in EEG, and in machine learning in general, are motivated precisely by data scarcity and generalisation issues. Without a clearer articulation of what makes their analysis novel in this space, the claim reads more as a restatement of standard goals rather than a meaningful point of differentiation.

- The claim that LaBraM is a "representative example" of self-supervised learning approaches (line 93) is non-trivial and unsubstantiated. LaBraM is just one specific method, and the field of self-supervised learning in EEG is diverse and covers paradigms like contrastive learning, masked prediction and generative approaches. The authors provide no justification for why LaBraM adequately captures the range of behaviors or limitations of these different methods. This weakens the argument that the findings generalise to self-supervised learning as a whole. A claim of representativeness must be supported by either a taxonomy of the space or comparative results, neither of which are provided.

- The claim in line 239 (doubling participants would yield “roughly 1%” improvement) is presented without any accompanying justification, reference to specific experiments or quantitative derivation. The number appears arbitrary. Given the paper’s central argument about the value of participant diversity, unsupported numerical claims like this undermine the credibility of the analysis.

- The question "Can contemporary EEG data augmentations effectively increase performance across different data regimes?" is framed too broadly given the narrow scope of the study. The authors test augmentation methods that are all relatively standard (e.g., Gaussian noise, cropping). No principled selection is made and there is no attempt to explore the space of more recent or task-specific augmentations, nor any ablation or combination strategies. As such, the findings are limited to a small subset of possible techniques and cannot support general conclusions about the effectiveness of augmentations across EEG tasks or regimes. The question overstates what the study can validly answer.

- The paper poses the broad question of whether self-supervised pre-training can improve performance across different data regimes, but only evaluates this using a single pre-trained model, LaBraM. This is insufficient to support a general conclusion. Evaluating just one model, trained on one dataset and applying it to a few target conditions does not constitute a rigorous or comprehensive assessment. The conclusion lacks external validity and is not grounded in enough empirical breadth to answer the question as posed.

---

> ### Author Rebuttal · Authors · 2025-07-31
>
> Thank you for your review and comments. We appreciate your recognition of the value of our theoretical analysis and the usefulness of our experiments.
>
>
>
> **Clarification of differentiation to prior work on data augmentations**
>
> The point we were trying to make is not that others have not looked at augmentations in the context of limited data. Our work looks at two different dimensions of data scarcity: the number of participants, and the amount of data available for each participant. In contrast, the only previous piece of work systematically investigating the impact of augmentations in EEG has only considered a single dimension encapsulating the total amount of data available.
>
> Moreover, we would like to emphasise that we provide additional reasons why our work is different in lines 85-89: “Furthermore, our experiments were conducted on much larger datasets with over 1,000 participants each, allowing us to study the impact of augmentations across a broader range of data regimes. Finally, their experiments were limited to a single model per task, whereas we benchmarked all augmentations with different model architectures for each task and observed differential effects.”
>
>
>
> **Representativeness of selected model for self-supervised learning**
>
> We will rephrase this claim in the manuscript to make our intentions clearer. LaBraM provides an example of a state-of-the-art method for self-supervised pre-training in EEG-based machine learning.
>
> We would like to remark that the use of only one SSL method does not impact our analysis of research question Q4. Our research question is stated as follows: “Q4: Can self-supervised pre-training effectively increase performance across different data regimes, including limited participant counts or data per participant?” This does not require us to show that *all* self-supervised learning methods increase performance. We can answer this question by showing that at least one self-supervised learning approach works.
>
>
>
> **Evidence for claim in line 239 (doubling participants would yield “roughly 1%” improvement)**
>
> This claim refers to the observations stated earlier in the paragraph. Specifically, we refer to normality prediction on the TUAB dataset in Figure 2, where extrapolating the curve after 1600 participants, we would expect roughly a 1% improvement if the participant count was doubled for the TCN or mAtt model. The same is already observable (so we don’t have to extrapolate) when moving from 800 to 1600 participants. Finally, this can also be seen by comparing the top 2 rows in any of the plots in Figure 3. We will clarify this in the manuscript.
>
>
>
> **Motivation for selected augmentations**
>
> First we would like to clarify that Gaussian noise and cropping are not augmentations studied in our work. We studied AmplitudeScaling, FrequencyShift, and PhaseRandomization.
>
> As discussed in the related works section, our selection was informed by the systematic evaluation of EEG augmentations conducted by Rommel et al. In their benchmarks on two tasks (sleep staging and BCI motor imagery), PhaseRandomization (termed FTSurrogate) consistently emerged as the most effective augmentation. FrequencyShift showed small improvements when data was more limited. Inclusion of the FrequencyShift augmentation was further motivated, because differences in peak frequencies have been shown to be associated with demographic variables like age and sex [1, 2, 3], making it an interesting augmentation to study in the context of participant diversity. In addition, we also studied AmplitudeScaling, an augmentation introduced by Mohsenvand et al. in the context of a contrastive learning approach, but not evaluated in the review by Rommel et al.
>
> Importantly, Rommel et al. also reported that other augmentations (Gaussian noise, channel symmetry, channel shuffle, time masking, bandstop filtering) did not provide consistent and significant improvements in performance, which is why we did not include them in our work. The two augmentations SignFlip and TimeReverse showed some improvements in limited settings, but were not included in our study because they do not preserve the structure of EEG data. Finally, while ChannelDropout seemed to provide limited value in the BCI task, a closer look revealed that the dropout rate was tuned to 1 (i.e. dropout of all channels), hinting at an issue with the evaluation task or the training, rather than genuine augmentation benefit.
>
> We will rephrase the relevant part of the related work section in the camera-ready version of the manuscript to make the motivation for our choice of augmentation methods more clear.
>
> Taken together, and given the more comprehensive evaluation of a wide range of augmentations in prior work that informed our study, we believe the three selected augmentations provide an informative picture of contemporary EEG data augmentations. We would also like to add that we do not exclude the possibility that more effective augmentations could be developed as discussed and encouraged in the discussion section of our manuscript.
>
>
> [1] Aurlien, H et al. ‘EEG Background Activity Described by a Large Computerized Database’. Clinical Neurophysiology, 2004.
>
> [2] Chiang, A. K. I. et al. ‘Age Trends and Sex Differences of Alpha Rhythms Including Split Alpha Peaks’. Clinical Neurophysiology, 2011.
>
> [3] Cellier, Dillan et al. ‘The Development of Theta and Alpha Neural Oscillations from Ages 3 to 24 Years’. Developmental Cognitive Neuroscience, 2021.

---

> > ### Comment · Reviewer_dTAg · 2025-08-05
> >
> > Thanks for providing the rebuttal and addressing a lot of my concerns. I have now raised my recommendation.

---

### Official Review · Reviewer_cReP · 2025-06-26

**Clarity:** 3
**Significance:** 3
**Originality:** 2
**Rating:** 4
**Confidence:** 4

**Summary:**

In this article, the authors investigate scaling laws for EEG-based decoding of normality prediction, dementia diagnosis, and sleep staging. In particular, they investigate how decoding performance scales with the number of training subjects and with the number of examples per subject. Additionally, they investigate if data augmentation techniques and SSL pre-training improve performance.

**Questions:**

Can you confirm that the examples were sampled using a uniform distribution over the available samples?

**Ethical Concerns:**

["NO or VERY MINOR ethics concerns only"]

**Final Justification:**

I expressed doubts regarding the methodology in my original review.
The authors provided us with an additional experiment with corrected methodology. It showed that this apparent gap did not impact the results, and the study conclusions remain unchanged.
Therefore, I increased my significance score (1 to 3) and my global score (2 to 4).

**Limitations:**

yes

**Quality:**

3

**Strengths And Weaknesses:**

The figures and diagrams are clear and of good quality (vectorized).
The research questions (RQs) are relevant. In particular, answers to RQ 2 (Is collecting more data per participant a reliable way to compensate for a lack of participants?) could be used to guide data collection, known to be costly, in the future.

However, **the experiments conducted by the authors do not allow to answer RQ2**. Indeed, to simulate a *scarce data regime*, the authors randomly sample examples from the train sets (lines 126 and 127). Because they do not provide more details, I assume that they are sampled using a uniform distribution over the available samples. This method is often used when benchmarking decoding algorithms. However, it can not be used to answer RQ 2 because it omits the long-term non-stationarities inherent to EEG recordings. These non-stationarities include slow drifts in the signal, loss of contact with the electrode leading to increased noise, subject fatigue, change in the mental state of the subject, eventual drug intake, etc. When the authors randomly sample n examples among all, they have a large diversity of examples regarding the different non-stationarities aforementioned, larger than if the n examples were consecutive, which will lead to overestimated generalization abilities.

Additionally, four RQs is probably too much content for one conference article. This article could easily be split into three separate articles: RQs 1+2, RQ3, and RQ4.
Finally, this article might out of the scope of NeurIPS as it does not introduce any novel method. However, with a corrected evaluation procedure to properly answer RQ 2, ths article would be of great interest for the BCI comunity.

---

> ### Author Rebuttal · Authors · 2025-07-31
>
> Thank you for your thoughtful review and feedback. We appreciate your acknowledgement of the relevance of our research questions and the practical value of our work for guiding data collection.
>
>
>
> **Uniform subsampling of segments**
>
> We confirm that segments were sampled uniformly in our experiments and we agree that contiguous sampling would more accurately reflect data collection in practice. Indeed, random subsampling of segments could lead to an artificially increased segment diversity and overestimation of model performance.
>
> To address this issue, we conducted additional experiments where segments are subsampled contiguously. Specifically, we repeated the experiments for the leftmost heatmap in Figure 3 (varying the number of participants and segments for normality prediction on TUAB with the TCN model) with segments sampled contiguously. We repeated these experiments for 5 different seeds. The results show that our conclusions still hold. In particular, even when segments are sampled contiguously instead of uniformly, collecting additional segments is not a reliable way to compensate for a lack of participants.
>
> We will include the results of this control experiment in the manuscript, showing accuracy heatmaps (with the standard error of the mean across seeds) for uniformly sampled vs contiguously sampled segments side-by-side. Since we are unable to share the plots via openreview, we provide the results in the form of a table here.
>
>
>
> Uniform sampling (average accuracies along with the standard error of the mean across 5 seeds):
>
> | n_participants \ n_segments | 5             | 10            | 20            | 40            | 80            | 160           | 320           |
> | --------------------------- | ------------- | ------------- | ------------- | ------------- | ------------- | ------------- | ------------- |
> | 25                          | 0.76 (0.0252) | 0.75 (0.0184) | 0.75 (0.0218) | 0.75 (0.0200) | 0.76 (0.0150) | 0.74 (0.0256) | 0.74 (0.0262) |
> | 50                          | 0.78 (0.0125) | 0.78 (0.0151) | 0.78 (0.0142) | 0.76 (0.0119) | 0.76 (0.0153) | 0.74 (0.0114) | 0.74 (0.0100) |
> | 100                         | 0.79 (0.0057) | 0.80 (0.0018) | 0.78 (0.0194) | 0.80 (0.0109) | 0.77 (0.0155) | 0.79 (0.0101) | 0.78 (0.0119) |
> | 200                         | 0.84 (0.0054) | 0.83 (0.0073) | 0.83 (0.0099) | 0.83 (0.0145) | 0.83 (0.0092) | 0.82 (0.0162) | 0.82 (0.0112) |
> | 400                         | 0.85 (0.0060) | 0.84 (0.0011) | 0.81 (0.0165) | 0.84 (0.0050) | 0.84 (0.0091) | 0.84 (0.0082) | 0.85 (0.0043) |
> | 800                         | 0.83 (0.0128) | 0.84 (0.0078) | 0.85 (0.0022) | 0.85 (0.0037) | 0.85 (0.0075) | 0.85 (0.0067) | 0.83 (0.0075) |
> | 1600                        | 0.83 (0.0117) | 0.84 (0.0056) | 0.84 (0.0058) | 0.85 (0.0046) | 0.85 (0.0030) | 0.85 (0.0029) | 0.84 (0.0054) |
>
>
>
> Contiguous sampling (average accuracies along with the standard error of the mean across 5 seeds):
>
> | n_participants \ n_segments | 5             | 10            | 20            | 40            | 80            | 160           | 320           |
> | --------------------------- | ------------- | ------------- | ------------- | ------------- | ------------- | ------------- | ------------- |
> | 25                          | 0.75 (0.0200) | 0.76 (0.0082) | 0.77 (0.0118) | 0.75 (0.0169) | 0.74 (0.0196) | 0.76 (0.0094) | 0.71 (0.0186) |
> | 50                          | 0.74 (0.0149) | 0.76 (0.0098) | 0.74 (0.0128) | 0.74 (0.0167) | 0.74 (0.0154) | 0.74 (0.0126) | 0.75 (0.0210) |
> | 100                         | 0.77 (0.0192) | 0.77 (0.0129) | 0.78 (0.0121) | 0.77 (0.0126) | 0.75 (0.0266) | 0.76 (0.0206) | 0.77 (0.0102) |
> | 200                         | 0.80 (0.0082) | 0.81 (0.0129) | 0.83 (0.0078) | 0.81 (0.0062) | 0.80 (0.0134) | 0.82 (0.0086) | 0.82 (0.0087) |
> | 400                         | 0.80 (0.0154) | 0.82 (0.0116) | 0.83 (0.0059) | 0.80 (0.0139) | 0.82 (0.0080) | 0.83 (0.0031) | 0.84 (0.0060) |
> | 800                         | 0.83 (0.0087) | 0.83 (0.0164) | 0.85 (0.0037) | 0.84 (0.0049) | 0.85 (0.0049) | 0.83 (0.0120) | 0.84 (0.0062) |
> | 1600                        | 0.84 (0.0039) | 0.83 (0.0134) | 0.85 (0.0014) | 0.84 (0.0104) | 0.85 (0.0029) | 0.85 (0.0018) | 0.85 (0.0025) |
>
>
>
> **Too much content**
>
> We have considered splitting the research questions into separate pieces of work. However, we believe that it is valuable to present them together under the theoretical framework introduced in Section 3. By including research questions 3 and 4, investigating the effectiveness of common machine learning methods to address data scarcity, we are able to present a more complete picture of current data limitations in EEG-based machine learning.
>
>
>
> **No novel methods**
>
> The NeurIPS2025 Reviewer Guidelines state that: “[...] originality does not necessarily require introducing an entirely new method. Rather, a work that provides novel insights by evaluating existing methods, or demonstrates improved efficiency, fairness, etc. is also equally valuable.”
>
> We believe it is fair to claim this for our work.

---

> > ### Comment · Reviewer_cReP · 2025-08-01
> >
> > Dear Authors,
> >
> > **Uniform subsampling of segments**
> >
> > Thank you for answering my points and for conducting this additional experiment. The new results, with "continuous sampling", are interesting and it is necessary to include them in the final manuscript, whether you publish it at NeurIPS or elsewhere.
> >
> > Could you provide a complete and precise description of your methodology for "continuous sampling"? I would be willing to increase my score if the sampling method is satisfactory.
> >
> > **Too much content**
> >
> > I still belives that four research questions is too ambitious for one conference article. As a result, some are not answered with enough depth. Indeed, reviewer LnQb misses content redarding the data augmentations, while reviewer 2Wvo misses a proper conclusion section and finds the discussion section limited. Finally, reviewer dTAg finds RQ 3 and 4 too broad and is not satisfied by the answers.
> >
> > **No novel methods**
> >
> > Thank you for this reminder

---

> > > ### Author Response · Authors · 2025-08-01
> > >
> > > Thank you for your reply. We sample contiguous segments as follows:
> > >
> > > We first divide an EEG recording into non-overlapping segments of fixed length. Given $N$ total segments in the recording and a desired number of segments $n_{\text{segments}}$, we uniformly sample a random starting index $s \in [0, N - n_{\text{segments}}]$. We then extract a contiguous block of $n_{\text{segments}}$ segments: $\text{segments}[s : s + n_{\text{segments}}]$.
> > >
> > > This ensures that the sampled segments are temporally contiguous within the original recording.

---

> > > > ### Comment · Reviewer_cReP · 2025-08-04
> > > >
> > > > Thank you for this clarification, it should be included in the final manuscript along with the additional experiment.
> > > >
> > > > I updated my review

---

### Official Review · Reviewer_LnQb · 2025-06-27

**Clarity:** 4
**Significance:** 3
**Originality:** 2
**Rating:** 5
**Confidence:** 3

**Summary:**

This paper examines the impact of participant diversity and data volume on the performance of machine learning models used for EEG analysis. The authors use various EEG datasets and tasks and analyze the effectiveness of data augmentation and self-supervised learning as strategies to address data limitations. Their findings suggest that participant diversity (i.e., in 2/3 tasks including more data per participant was not effective to compensate for small participant numbers) often limits model generalization, highlighting the need for methods that specifically tackle inter-individual distribution shifts.

**Questions:**

1. What was the train/test/validation split used for each dataset?
2. The paper shows that common EEG data augmentation methods failed to improve performance and, in some cases, even decreased the performance. While the authors mention that PhaseRandomization can destroy wave morphology and cross-channel correlation, which are critical for sleep stages, could the authors elaborate why they believe that AmplitudeScaling and FrequencyShift did not show an improvement in accuracy? Additionally,  what type of augmentations would mitigate participant differences?


Smaller improvement:
- Figure 5 caption: Capitalise pre-training. The sentence should be: "...baseline models for the same seed. Pre-training consistently improves..."

**Ethical Concerns:**

["NO or VERY MINOR ethics concerns only"]

**Final Justification:**

In the rebuttal, the authors cleared up my concerns, and I appreciate the effort. They also responded thoughtfully to all the reviewers and covered most of the concerns that were raised. After reading the discussions and answers provided to the other reviewers, I agree with some of the concerns that the paper is not methodologically novel, but I think the paper provides novel insights, so I've bumped my score up from a 4 to a 5.

**Limitations:**

The paper includes a dedicated section for "Limitations and Future Work" in Appendix A.1, however for transparency, this should be moved to the main text.

**Paper Formatting Concerns:**

The paper includes a dedicated section for "Limitations and Future Work" in Appendix A.1

**Quality:**

3

**Strengths And Weaknesses:**

Strengths:
- The authors distinguish between overall sample size and participant diversity (i.e., number of participants and segments per participant). This level of experimental control enables a careful disentanglement of their respective effects.
- The paper is clearly written and well-structured.
- The study presents a systematic comparison across models and datasets. The use of multiple datasets and model types strengthens the generalizability of the findings. It is particularly valuable that the authors directly compare the effects of self-supervised pre-training and data augmentation.

Weaknesses
- The discussion of limitations is confined to the appendix. These should be addressed in the main text to provide a more balanced and transparent assessment of the work.

---

> ### Author Rebuttal · Authors · 2025-07-31
>
> Thank you for your constructive review and insights. We appreciate your recognition of the generalisability of our findings and the value of directly comparing the effects of data augmentations and self-supervised pre-training.
>
>
>
> **Limitations section in the appendix**
>
> The camera ready copy allows for an additional page in the main paper, in which we will include this limitations section.
>
>
>
> **Details on train/val/test splits**
>
> The official train-test splits were used for TUAB and CAUEEG, and in both cases part of the train split was set aside for validation. For PhysioNet, where labels for the official test split are not publicly available, we created an age and sex stratified split into train, val, and test on the participant level (i.e. without overlapping participants). We will clarify this in the manuscript. For all datasets, the exact splits we used can be reproduced with the shared code.
>
>
>
> **Discussion of AmplitudeScaling and FrequencyShift augmentations**
>
> We will add the following discussion to address why AmplitudeScaling and FrequencyShift may not have improved model performance:
>
> > A likely explanation for why AmplitudeScaling and FrequencyShift did not yield consistent accuracy improvements is that these augmentations are not truly label preserving. For example, overall amplitude and differences in amplitude between EEG channels (e.g. asymmetries between corresponding channels on the left and right hemisphere) can provide clinically relevant information [1]. Similarly, frequency shifts can be clinically relevant (e.g. slowing may reflect abnormalities [1] and has been linked to MCI and AD [2, 3]). Alternatively, it is possible that even the smallest amount of training data already provided sufficient diversity of amplitudes and frequency shifts for the model to learn sufficient invariances to these aspects of the data, rendering the augmentations redundant.
> >
> > [1] Rubin, Devon I. et al. ‘Clinical Neurophysiology’. Contemporary Neurology Ser., 2021.
> >
> > [2] Cassani, R. et al. ‘Systematic Review on Resting-State EEG for Alzheimer’s Disease Diagnosis and Progression Assessment’. Disease Markers, 2018.
> >
> > [3] Modir, Aslan et al. ‘A Systematic Review and Methodological Analysis of EEG-Based Biomarkers of Alzheimer’s Disease’. Measurement, 2023.
>
>
>
> **Discussion of potential augmentations to mitigate participant differences**
>
> What augmentations would effectively simulate participant differences is an open research question for which we encourage future work in our discussion section. An important consideration will be the nature of participant distribution shifts as we outline in the Future Work section: for example, if we hypothesise that anatomical differences are the main driver of participant diversity, future work could attempt to develop augmentations to imitate anatomical differences such as skull thickness or head circumference. If participant distribution shifts are instead explained by variability in disease phenotypes, domain knowledge of the specific disease could inform the design of augmentations for it.

---

> > ### Comment · Reviewer_LnQb · 2025-08-06
> >
> > In the rebuttal, the authors cleared up my concerns, and I appreciate the effort. They also responded thoughtfully to all the reviewers and covered everything that was raised. After reading the discussions and answers provided to the other reviewers, I agree with some of the concerns that the paper is not methodologically novel, but I think the paper provides novel insights, so I've bumped my score up from a 4 to a 5.

---

### Official Review · Reviewer_77HZ · 2025-06-27

**Clarity:** 3
**Significance:** 2
**Originality:** 2
**Rating:** 3
**Confidence:** 3

**Summary:**

This paper explores the impact of both sample size and participant diversity for model generalisability and robustness when applying machine learning to EEG data. By viewing EEG generation as a multi-level process, the authors empirically analyse how model performance scales with data quantity and participant diversity. They further evaluate the impact of data augmentation and self-supervised learning in scenarios with limited data. This study reveals that distributional shifts among participants can significantly limit performance gains and offers practical recommendations for data collection and machine learning methodologies. The experimental code is also made publicly accessible to advance research within the community.

**Questions:**

1. The main answers to the proposed four questions in Section 4 seem quite intuitive and might not offer much novelty to readers already knowledgeable in this area. Could the authors clarify what new insights or conceptual advancements their work brings beyond what is already expected?

2. The purpose and significance of the analysis presented in Section 3 are somewhat ambiguous. How does this section specifically relate to the core experimental results or support the main conclusions of the paper? Providing a clearer explanation of its role would help improve the coherence and impact of this study.

**Ethical Concerns:**

["NO or VERY MINOR ethics concerns only"]

**Final Justification:**

The authors have provided a thoughtful rebuttal addressing the main concerns.

Regarding the concern about the intuitive nature of the conclusions, the authors clarified that some of their findings—such as the failure of data augmentation on large-scale EEG datasets and the dominant role of participant diversity—challenge common assumptions and offer actionable insights. Their recommendation for pilot studies to assess participant diversity is well-argued.

The concern regarding the role of Section 3 was also addressed with a clearer explanation of how the theoretical framework supports the experimental results and research questions.

However, I believe the paper would benefit further from developing a more explicit or trade-off framework that quantifies the optimal balance between the number of participants and the number of samples per participant. Such a formulation could serve as a practical tool for dataset construction and selection in EEG and ML research. Given that data scale, diversity, and domain alignment between pretraining and downstream datasets are all critical to model training and generalization, formalizing this interplay would significantly enhance the utility and impact of the findings.

Moreover, while the paper frames itself as an empirical study of scaling behaviour and participant diversity in EEG modeling, the experimental coverage remains somewhat limited. Only three datasets (TUAB, CAUEEG, PhysioNet), three models (TCN, mAtt, LaBraM), and three task types (normality prediction, dementia diagnosis, sleep staging) were included. For a study aiming to establish broadly applicable empirical trends, this setup may be too narrow to support strong general conclusions. Including a wider range of EEG tasks, datasets with varying recording protocols, and more diverse model architectures would substantially strengthen the work’s external validity and practical relevance.

Overall, while the paper is methodologically sound and the rebuttal addresses the raised concerns, I still find the broader impact to be somewhat limited.

**Limitations:**

Yes.

**Paper Formatting Concerns:**

None.

**Quality:**

2

**Strengths And Weaknesses:**

**Strengths**:

1.	The paper presents what appears to be the first systematic investigation into how the number of data samples and subjects influences the performance of EEG-based machine learning models.
2.	The experimental design is thorough and methodologically sound, incorporating both supervised and self-supervised learning approaches across datasets from diverse domains.

**Weaknesses**:

1.	The main conclusions—particularly the answers to the four questions posed in Section 4—are largely intuitive and may be considered unsurprising by readers familiar with the field. As a result, the paper offers limited new insights or conceptual advancement.
2.	The role of the analysis in Section 3 is unclear. It is not evident how this section connects with the core experiments or contributes directly to the study’s central conclusions. A more explicit explanation of its relevance would strengthen the narrative.

---

> ### Author Rebuttal · Authors · 2025-07-31
>
> Thank you for your helpful review and comments. We appreciate that you found our experiments thorough and methodologically sound.
>
>
>
> **Limited new insights due to intuitive conclusions**
>
> First, we would like to emphasise that our results provide quantitative, more nuanced, surprising, and non-trivial insights.
>
> - We evaluated EEG data augmentations on much larger datasets and across a more diverse set of models and tasks than prior work. Our results show that data augmentations largely failed to improve performance—a conclusion that stands in contrast to their reported success in smaller-scale studies in the literature.
> - Our experiments provide *quantitative* evaluations of the scaling behaviour of different models and the impact of participant diversity across multiple tasks. We would argue that it was not expected that even a more than 50-fold increase in sample size would fail to improve performance in the absence of sufficient participant diversity on TUAB. This result demonstrates how severely model performance can be bottlenecked by the participant count.
> - It is currently not common practice to inform data collection with pilot studies that evaluate the importance of participant diversity. As mentioned in the discussion section, our results motivate such pilot studies given the severely participant-count-dominated scaling behaviour on TUAB and CAUEEG on the one hand, and the PhysioNet results on the other hand, where participant diversity was less critical. We are not aware of prior work that provides such actionable guidance using empirical scaling results.
>
> Finally, while some of our results might be well aligned with the intuition of experts in the field, we argue that it is important to test intuitions and substantiate them with empirical results. When two researchers have conflicting intuitions about the existence or importance of participant distribution shifts, our work provides empirical evidence that can inform the discussion.
>
>
>
> **Role of the analysis in Section 3**
>
> In Section 3, we lay out the theoretical foundation that motivates our experiments and provides a framework in which the results can be interpreted.
>
> Section 3 currently concludes with the following summary:
>
> > “This analysis tells us that in the presence of a substantial distribution shift, the extent to which one should expect to overfit is governed primarily by the number of participants, rather than the amount of data available per participant, unless one is able to design a model architecture with the appropriate invariances.”
>
> To make the connection to our research questions clear, we will extend this paragraph with the following explanation:
>
> > “Under this framework, we address the four research questions outlined in the introduction. We assess the relevance and magnitude of participant distribution shifts through the scaling behaviour of model performance (Q1 & Q2) and the effectiveness of augmentations (Q3) and self-supervised pre-training (Q4) to improve performance by endowing models with useful invariances or boosting effective participant counts.”

---

> ### Comment · Reviewer_cReP · 2025-08-01
>
> Dear reviewer 77HZ,
>
> Please note that this manuscript is not the first to explore how the performance of EEG-based machine learning models is influenced by number of subjects and number of trials. See Banville et. al (2025) 	https://arxiv.org/abs/2501.15322

---

> > ### Author Response · Authors · 2025-08-01
> >
> > We discuss this work in our related works section: A critical difference to our work is that in their study on decoding images form brain activity, data from the same participant is used during training and test time, i.e. models do not have to generalise to unseen participants. Our work is focused on setting where generalisation to new participants is crucial, including applications like diagnosis or prognosis.

---

> > ### Comment · Reviewer_77HZ · 2025-08-05
> >
> > Dear Reviewer cReP,
> >
> > Thank you so much for clarifying this point.

---

> ### Comment · Reviewer_77HZ · 2025-08-05
>
> Thank you for your efforts in the rebuttal. I appreciate the time and thought you put into addressing the comments.

---

### Official Review · Reviewer_2Wvo · 2025-06-29

**Clarity:** 3
**Significance:** 3
**Originality:** 3
**Rating:** 4
**Confidence:** 3

**Summary:**

This paper systematically investigates the impact of participant diversity versus overall sample size on the performance of EEG-based machine learning models. The findings highlight that participant diversity can be a critical factor, often more so than simply increasing the total number of segments, and demonstrate the potential of self-supervised pre-training to mitigate data scarcity.

**Questions:**

1. Why were TCN, mAttn and LaBraM chosen? Can these three models cover the commonly used method categories in EEG-based tasks? It is suggested to explain it in the paper.
2. In each task domain (normality prediction, dementia diagnosis, sleep staging), what augmentations are commonly adopted in prior literature? Are amplitude scaling, frequency shifting and phase randomization considered best-practice baselines? If not, how can the universality of your Q3 conclusion be claimed?

**Ethical Concerns:**

["NO or VERY MINOR ethics concerns only"]

**Final Justification:**

During the rebuttal session, the authors have addressed most of my concerns, and I appreciate the clarification regarding the motivation for the choice of augmentation methods, which will be rephrased in the camera-ready version. However, I think the methodological novelty is somewhat limited. Therefore, I have increased my global score from 3 to 4.

**Limitations:**

YES

**Quality:**

3

**Strengths And Weaknesses:**

# Strength
1. Good writing and organization
2. The paper provides a valuable and systematic empirical analysis of the relative importance of "participant diversity" and "overall sample size" in EEG machine learning, a crucial aspect often overlooked in prior work.
# Weakness
1. The experiments in this paper are not sufficient the number of tasks, models, and datasets used is too small, which reduces the persuasiveness of the results.
2. The manuscript lacks a conclusion section.
3. While the manuscript provides a detailed analysis of the experimental phenomena, it offers limited discussion on the underlying causes behind these observations.

---

> ### Author Rebuttal · Authors · 2025-07-31
>
> Thank you for your review and constructive feedback. We are glad you valued our systematic empirical analysis.
>
>
>
> **Limited number of tasks, models, and datasets**
>
> We aimed to design a diverse and sufficiently broad set of conditions to support meaningful and generalisable conclusions. Our experimental setup provides a solid basis to answer our research questions, and experiments on additional datasets would not change our conclusions.
>
> - We deliberately included three models with fundamentally different architectures: TCN (CNN-based), mAtt (geometric deep learning using covariance representations), and LaBraM (Transformer-based). These span a range of methodological paradigms explored in recent EEG literature and reduce the likelihood of model-specific conclusions.
> - We focused on large-scale datasets (1000+ participants), which are rare in EEG research [1]. Large datasets are essential for our experimental design, as they allow us to systematically vary the number of participants and segments over several orders of magnitude. Our work could not have been done with a large number of small datasets. Additionally, evaluations on small datasets, while easier to scale, are often noisy and less reliable.
> - Compared to prior work, we are not aware of any EEG-based ML study with evaluations on a higher number of large (1000+ participants) datasets, diverse model architectures, and different tasks.
> - Additional datasets or tasks would not change the answers to our research questions. Normality prediction on TUAB and dementia diagnosis on CAUEEG already provide striking examples of severe participant distribution shifts, whereas sleep staging experiments on PhysioNet showed that there are specific applications where it is possible to achieve good performance even with limited participant counts. So while additional tasks and datasets could have some value to inform application-specific future work, we do not require more experiments to answer the research questions in this paper.
>
> Finally, we would like to emphasise the computational scale of our current experiments. For example, for the baseline experiments on TUAB alone, we evaluated 49 different data regimes (combinations of participant and segment counts) for each of the three models and repeated each configuration with 25 random seeds, resulting in 3675 training runs.
>
>
>
> [1] Roy, Yannick et al. ‘Deep Learning-Based Electroencephalography Analysis: A Systematic Review’. Journal of Neural Engineering, 2019.
>
>
>
> **Missing conclusion section**
>
> We aimed to use the Discussion section as an extended Conclusion section. We will include a short summary of the main points of this Discussion section in a new Conclusion section for the camera ready version.
>
>
>
> **More discussion of underlying causes behind observations**
>
> - Observations for Q1 and Q2 are addressed through the analysis in Section 3, which describes a theoretical framework under which the importance of participant diversity can be explained. We link back to this analysis in the results section, e.g. line 264.
> - Observations for Q3 are discussed in Section 4.3. For instance, we discuss underlying reasons why the PhaseRandomization augmentation failed to improve performance in line 291. We will extend the camera-ready version of the manuscript with further discussion about AmplitudeScaling and FrequencyShift (see below).
> - Observations for Q4 are discussed in Section 4.4. We will extend the camera-ready version of the manuscript with further discussion (see below).
> - In addition, model differences are discussed in a dedicated section in A.5 with references and additional discussion in the results section, e.g. lines 248 and 287.
>
>
>
> Extended discussion of observations about AmplitudeScaling and FrequencyShift:
>
> > A likely explanation for why AmplitudeScaling and FrequencyShift did not yield consistent accuracy improvements is that these augmentations are not truly label preserving. For example, overall amplitude and differences in amplitude between EEG channels (e.g. asymmetries between corresponding channels on the left and right hemisphere) can provide clinically relevant information [1]. Similarly, frequency shifts can be clinically relevant (e.g. slowing may reflect abnormalities [1] and has been linked to MCI and AD [2, 3]). Alternatively, it is possible that even the smallest amount of training data already provided sufficient diversity of amplitudes and frequency shifts for the model to learn sufficient invariances to these aspects of the data, rendering the augmentations redundant.
> >
> > [1] Rubin, Devon I. et al. ‘Clinical Neurophysiology’. Contemporary Neurology Ser., 2021.
> >
> > [2] Cassani, R. et al. ‘Systematic Review on Resting-State EEG for Alzheimer’s Disease Diagnosis and Progression Assessment’. Disease Markers, 2018.
> >
> > [3] Modir, Aslan et al. ‘A Systematic Review and Methodological Analysis of EEG-Based Biomarkers of Alzheimer’s Disease’. Measurement, 2023.
>
>
>
> Extended discussion of observations for self-supervised pre-training:
>
> > Pre-training exposed the model to a larger diversity of EEG data from additional participants than those seen during fine-tuning. The positive results suggest that, even without labels, this has allowed the model to learn useful representations that transfer to diverse downstream tasks.
>
>
>
> If there are other observations that the reviewer believes would benefit from further discussion, could they let us know more specifically what they are?
>
> **Motivation for selected models**
>
> We selected three influential and widely used models with distinct architectures to obtain a diverse perspective on EEG model behavior and reduce the risk of model-specific effects.
>
> - **TCN** is a convolutional neural network (CNN), which was developed for generic sequence modeling and has since been widely adopted in EEG research. CNNs remain a strong baseline in EEG-based machine learning.
> - **mAtt** is a geometric deep learning model specifically developed for EEG. It encodes signals as sequences of covariance matrices and exemplifies a line of work that incorporates Riemannian geometry to leverage domain-specific structure.
> - **LaBraM** is a transformer architecture. As in other domains, transformers have become a common architecture choice in recent works, particularly for the development of foundation models.
>
> These models provide a good picture of the landscape of the current state-of-the-art in EEG-based machine learning.
>
>
>
> **Motivation for selected augmentations**
>
> As discussed in the related works section, our selection was informed by the systematic evaluation of EEG augmentations conducted by Rommel et al. In their benchmarks on two tasks (sleep staging and BCI motor imagery), PhaseRandomization (termed FTSurrogate) consistently emerged as the most effective augmentation. FrequencyShift showed small improvements when data was more limited. Inclusion of the FrequencyShift augmentation was further motivated, because differences in peak frequencies have been shown to be associated with demographic variables like age and sex [1, 2, 3], making it an interesting augmentation to study in the context of participant diversity. In addition, we also studied AmplitudeScaling, an augmentation introduced by Mohsenvand et al. in the context of a contrastive learning approach, but not evaluated in the review by Rommel et al.
>
> Importantly, Rommel et al. also reported that other augmentations (Gaussian noise, channel symmetry, channel shuffle, time masking, bandstop filtering) did not provide consistent and significant improvements in performance, which is why we did not include them in our work. The two augmentations SignFlip and TimeReverse showed some improvements in limited settings, but were not included in our study because they do not preserve the structure of EEG data. Finally, while ChannelDropout seemed to provide limited value in the BCI task, a closer look revealed that the dropout rate was tuned to 1 (i.e. dropout of all channels), hinting at an issue with the evaluation task or the training, rather than genuine augmentation benefit.
>
> We will rephrase the relevant part of the related work section in the camera-ready version of the manuscript to make the motivation for our choice of augmentation methods more clear.
>
> Taken together, and given the more comprehensive evaluation of a wide range of augmentations in prior work that informed our study, we believe the three selected augmentations provide an informative picture of contemporary EEG data augmentations. We would also like to add that we do not exclude the possibility that more effective augmentations could be developed as discussed and encouraged in the discussion section of our manuscript.
>
> [1] Aurlien, H et al. ‘EEG Background Activity Described by a Large Computerized Database’. Clinical Neurophysiology, 2004.
>
> [2] Chiang, A. K. I. et al. ‘Age Trends and Sex Differences of Alpha Rhythms Including Split Alpha Peaks’. Clinical Neurophysiology, 2011.
>
> [3] Cellier, Dillan et al. ‘The Development of Theta and Alpha Neural Oscillations from Ages 3 to 24 Years’. Developmental Cognitive Neuroscience, 2021.

---

> > ### Comment · Reviewer_2Wvo · 2025-08-04
> >
> > Thanks for the clarifications. The authors have solved most of my confusion and questions.  I will increase my score.

---

### Decision · Program_Chairs · 2025-09-17

**Decision:**

Accept (poster)

**Comment:**

This work proposes an in depth and well executed study of the impact of evaluating ML model on neural EEG data from novel subjects (can be seen as out of domain generalisation where new subjects are new domains). Reviewers have appreciated the detailed rebuttal and consider their concerns addressed. While the ML contribution can be judged limited for NeurIPS, the detailed discussion and new insights are considered interesting to move the field forward, justifying an endorsement for publication.